# Learning Entity Representations for Few-Shot Reconstruction of Wikipedia Categories

## Abstract

Language modeling tasks, in which words are predicted on the basis of a local context, have been very effective for learning word embeddings and context dependent representations of phrases. Motivated by the observation that efforts to code world knowledge into machine readable knowledge bases tend to be entity-centric, we investigate the use of a fill-in-the-blank task to learn context independent representations of entities from the contexts in which those entities were mentioned. We show that large scale training of neural models allows us to learn extremely high fidelity entity typing information, which we demonstrate with few-shot reconstruction of Wikipedia categories. Our learning approach is powerful enough to encode specialized topics such as *Giro d'Italia cyclists*.

## 1 Introduction

A long term goal of artificial intelligence has been the development and population of an entity-centric representation of human knowledge. Efforts have been made to create the knowledge representation with knowledge engineers (Lenat et al., 1986) or crowdsourcers (Bollacker et al., 2008). However, these methods have relied heavily on human definitions of their ontologies, which are both limited in scope and brittle in nature. Conversely, due to recent advances in deep learning, we can now learn robust general purpose representations of words (Mikolov et al., 2013) and contextualized phrases (Peters et al., 2018; Devlin et al., 2018) directly from large textual corpora.

Consider the following *context* in which an entity mention is replaced with the [MASK] symbol:

> ... [MASK], a Russian factory worker, was the first woman in space ...

As readers, we understand that *first woman in space* is a unique identifier, and we are able to fill in the blank unambiguously. The central hypothesis of this paper is that, by matching entities to the contexts in which they are mentioned, we should be able to build a representation for Valentina Tereshkova that encodes the fact that she was the first woman in space.

To do this, we start with BERT (Devlin et al., 2018), a powerful pretrained text encoder, to encode contexts—Wikipedia text in which a hyperlinked span has been blanked out—and we train an entity encoder to match the BERT representation of the entity's contexts. We experiment with a lookup table that maps each entity to a fixed length vector, which we call **RELIC** (Representations of Entities Learned In Context). We hypothesize that the dedicated entity representations in RELIC should be able to capture knowledge that is not present in BERT. To test this, we compare RELIC to two BERT-based entity encoders: one that encodes the entity's canonical name, and one that encodes the first paragraph of the entity's Wikipedia page.

Ultimately, we would like our representations to encode all of the salient information about each entity. However, for this initial work, we study our representations' ability to capture Wikipedia categorical information encoded by human experts. We show that given just a few exemplar entities of a Wikipedia category such as *Giro d'Italia cyclists*, we can use RELIC to recover the remaining entities of that category with good precision.

## 2 Related work

Several works have tackled the problem of learning distributed representations of entities in a knowledge base (KB). Typical approaches rely on the (subject, relation, object) ontology of KBs like Free-

base (Bollacker et al., 2008). These methods embed the entities and relations in vector space, then maximize the score of observed triples against negative triples to do KB completion (Bordes et al., 2011; Socher et al., 2013; Yang et al., 2014; Toutanova et al., 2016).

There is relatively less work in learning entity representations directly from text. The contextual approach of word2vec (Mikolov et al., 2013) has been applied to entities, but there has been little analysis of how effective such a method would be for answering the entity typing queries we study in this work. Most methods for entity representations that do use raw text will combine it with structure from an existing KB (Riedel et al., 2013; Toutanova et al., 2015; Hu et al., 2015) in an effort to leverage as much information as possible. While there may be gains to be had from using structure, our goal in this work is to isolate and understand the limits of inducing entity representations from raw text alone.

We also note the similarity of our RELIC task to entity linking (Ratinov et al., 2011; Cheng & Roth, 2013; Sun et al., 2015; Gupta et al., 2017) and entity typing (Yaghoobzadeh & Schütze, 2015; Shimaoka et al., 2016), where entity mentions are processed in context. Unlike previous work in context-dependent entity typing (Ling & Weld, 2012; Choi et al., 2018; Murty et al., 2018), we consider types of RELIC from a *global* perspective. We are interested in identifying context-independent types of entities so that they can be used to identify structure in the entity latent space.

## 3   LEARNING FROM CONTEXT

We study the ability of current models to learn entity encoders directly from the contexts in which those entities are seen. Formally, we define an entity encoder to be a function $\mathbf{h}_e = f(e)$ that maps each entity $e$ to a vector $\mathbf{h}_e \in \mathbb{R}^d$. We outline the training procedure used to learn the encoders.

**RELIC training input**   Let $\mathcal{E} = \{e_0 \ldots e_N\}$ be a predefined set of entities, and let $\mathcal{V} = \{[\text{MASK}], w_1 \ldots w_M\}$ be a vocabulary of words. A *context* $\mathbf{x} = [x_0 \ldots x_l]$ is a sequence of words $x_i \in \mathcal{V}$. Each context contains exactly one instance of the [MASK] symbol. Our training data is a corpus of (context, entity) pairs $\mathcal{D} = [(\mathbf{x}_0, y_0) \ldots (\mathbf{x}_N, y_N)]$. Each $y_i \in \mathcal{E}$ is an entity, and the [MASK] symbol in $\mathbf{x}_i$ substitutes for a single mention $y_i$. For clean training data, we extract our corpus from English Wikipedia, taking advantage of its rich hyperlink structure (Section 3.2).

**RELIC training loss**   We introduce a context encoder $\mathbf{h}_\mathbf{x} = g(\mathbf{x})$ that maps the context $\mathbf{x}$ into the same space $\mathbb{R}^d$ as our entity encodings. Then we define a compatibility score between the entity $e$ and the context $\mathbf{x}$ as the scaled cosine similarity $s(\mathbf{x}, e) = a \cdot \frac{g(\mathbf{x})^\top f(e)}{||g(\mathbf{x})|| ||f(e)||}$ where the scaling factor $a$ is a learned parameter, following Wang et al. (2018). Now, given a context $\mathbf{x}$, the conditional probability that $e$ was the entity seen with $\mathbf{x}$ is defined as $p(e|\mathbf{x}) = \frac{\exp(s(\mathbf{x},e))}{\sum_{e' \in \mathcal{E}} \exp(s(\mathbf{x},e'))}$ and we train RELIC by maximizing the average log probability $\frac{1}{|\mathcal{D}|} \sum_{(\mathbf{x},y) \in \mathcal{D}} \log p(y|\mathbf{x})$.

In practice, we use a noise contrastive loss (Gutmann & Hyvärinen, 2012; Mnih & Kavukcuoglu, 2013), where we sample $K$ negative entities $e'_1, e'_2, \ldots, e'_K$ from a noise distribution $p_{noise}(e)$. Denoting $e'_0 := e$, our per-example loss is $l(s, \mathbf{x}, e) = -\log \frac{\exp(s(\mathbf{x},e))}{\sum_{j=0}^{K} \exp(s(\mathbf{x},e'_j))}$. We train our model with minibatch gradient descent and use all other entries in the batch as negatives. This is roughly equivalent to $p_{noise}(e)$ being proportional to entity frequency.

### 3.1   REPRESENTING ENTITIES AND CONTEXTS

**BERT context encoder**   For $g$, we use the pretrained BERT model (Devlin et al., 2018), a powerful Transformer-based (Vaswani et al., 2017) text encoder, to encode contexts into a fixed-size representation[1]. We project the BERT hidden state into $\mathbb{R}^d$ using a weight matrix $W \in \mathbb{R}^{d \times 768}$ to obtain our context encoding.

**RELIC entity encoder**   The entity encoder $f : \mathcal{E} \to \mathbb{R}^d$ maps symbolic entity representations into a vector space. We assume that each entity is identified by a unique Wikidata QID[2], a canonical

---

[1] We use lowercase BERT-Base (12 layers, 768 hidden size, 12 attention heads), and the initial hidden state of the Transformer (corresponding to the [CLS] token of BERT) as a fixed size representation.

[2] https://www.wikidata.org/wiki/Q43649390

| | | 3 exemplars | | | 10 exemplars | | |
|---|---|---|---|---|---|---|---|
| $K$ | $\bar{p}$ | RELIC | Name | Desc | RELIC | Name | Desc |
| | | All entities | | | | | |
| 10 | 1.00 | 70.64 | 86.63 | 76.76 | 73.56 | 87.35 | 81.83 |
| 1000 | 1.68 | 26.61 | 29.71 | 29.21 | 30.82 | 37.19 | 32.33 |
| All (4M) | 1848.15 | 7.67 | 9.54 | 12.02 | 10.24 | 11.42 | 13.78 |
| | | Entities seen at least 10 times in $\mathcal{D}$ | | | | | |
| 10 | 1.02 | 93.48 | 85.96 | 68.36 | 94.58 | 90.98 | 74.74 |
| 1000 | 2.78 | 43.39 | 32.95 | 17.90 | 52.04 | 44.27 | 22.98 |
| All (1M) | 2539.02 | 24.38 | 17.51 | 6.39 | 29.00 | 21.76 | 8.45 |

Table 1: Results for the Wikipedia category population task. Mean Average Precision for ranking entities given a set of exemplars of a given Wikipedia class. $K$ represents the number of candidates to be ranked, and $\bar{p}$ is the average number of positive labels among the candidates. Results are averaged over 100 categories sampled at random from those containing at least 1000 entities.

Wikipedia name, and the first paragraph of its Wikipedia page. We consider three encoders that operate on different representations of the entities: (1) embedding lookup, (2) BERT name encoder, and (3) BERT description encoder.

In the standard RELIC setup, we map each entity, identified by its unique ID, directly onto its own dedicated vector in $\mathbb{R}^d$ via a $|\mathcal{E}| \times d$ dimensional embedding matrix. We also consider two alternate BERT-based options for distributed encoding of entities, which are fine-tuned on the RELIC data. The *name encoder* applies a BERT Transformer to the canonical name of the entity to obtain a fixed-size representation. The *description encoder* applies a BERT Transformer to an entity's description to obtain a fixed size representation[3]. Note that both name and description encoders can do zero-shot encoding of new entities, assuming that a name or description is provided.

## 3.2 EXPERIMENTAL SETUP

To train RELIC, we obtain data from the 2018-10-22 dump of English Wikipedia. We take $\mathcal{E}$ to be the set of all entities in Wikipedia (of which there are over 5 million). For each hyperlink, we take the context as the surrounding sentence, replace all tokens in the anchor text with a single [MASK] symbol, and set the entity linked to as ground truth. We limit each context to 64 tokens.

We set the entity embedding size to $d = 300$. For the name and description encoders, we take the initial hidden state of the Transformer as the fixed-size representation. We limit to 16 name tokens and 128 description tokens. We train the model using TensorFlow (Abadi et al., 2016) with a batch size of 1024 for 5 epochs. We hold out about 1/30,000 of the data for use as validation, on which the final model achieves roughly 85% accuracy in-batch negative prediction for all models.

## 4 POPULATING WIKIPEDIA CATEGORIES

We introduce a fine-grained entity typing task based on Wikipedia categories, where the task is to populate a category from a small number of exemplars. We evaluate if RELIC benefits from dedicated embeddings over the BERT encoders that share parameters between entities.

We filter the Wikipedia categories in Yago 3.1 (Mahdisoltani et al., 2013) to get the 1,129 categories that cover at least 1,000 entities and consider an exemplar based "few-shot" scenario, based on the prototypical approach of Snell et al. (2017). For each category, we provide a small number of exemplars (3 or 10), one correct candidate entity drawn from the category, and $K-1$ other candidate entities. The candidate entities are ranked according to the inner product between their RELIC embeddings and the centroid of the exemplar embeddings, and we report the mean average precision (MAP) for entities belonging to the query class. Wikipedia categories are often incompletely labeled, and when $K$ covers all entities, this confounds the MAP calculation. Therefore, we also present results for $K = 10$ and $K = 1000$ for a cleaner experimental setup.

---

[3]We observe that while embeddings and name encoders could be applied to any entity dictionary, the description encoder may be less broadly applicable to other domains outside Wikipedia.

| Freq | # entities | RELIC | Name | Desc |
|---|---|---|---|---|
| $[0, 1)$ | 23k | 0.00 | 7.52 | **12.56** |
| $[1, 10)$ | 89k | 4.81 | 11.53 | **14.35** |
| $[10, 100)$ | 61k | **25.01** | 12.04 | 12.97 |
| $[100, 1k)$ | 11k | **18.18** | 5.83 | 8.90 |
| $[1k, 10k)$ | 650 | **4.93** | 1.66 | 0.06 |
| 10k+ | 10 | **0.08** | 0.00 | 0.00 |
| All | 184k | 7.67 | 9.54 | **12.02** |

Table 2: Mean Average Precision for 100 sampled Wikipedia categories, bucketed by how often each entity appears in our RELIC pretraining data.

| Category | Exemplars | Top 10 |
|---|---|---|
| Action films | (1) Oblivion (2) Blood Diamond (3) Fight Club | **(1) The Bourne Supremacy (2) The Bourne Identity** (3) Zodiac **(4) Collateral** (5) Signs **(6) 2012** (7) **The Island (8) The Bourne Ultimasty (9) I Am Legend (10) Live Free or Die Hard** |
| American Biologists | (1) Michael Murphy Andregg (2) Mark Peifer (3) Mary Jane West-Eberhard | (1) Warder Clyde Allee **(2) Richard Lewontin** (3) Lancelot Hogben **(4) William Earnest Castle (5) Emile Zuckerkandl** (6) Lawrence Joseph Henderson (7) Alfred J. Lotka (8) Herbert Spencer Jennings (9) G. Evelyn Hutchinson **(10) H.J. Muller** |
| Video games featuring female protagonists | (1) The Grinder (2) Final Fantasy XIII (3) Space Raiders | **(1) Final Fantasy X-2 (2) Final Fantasy X** (3) Final Fantasy Type-0 **(4) Final Fantasty XIII-2** (5) Final Fantasy XV (6) Final Fantasy XII (7) Tales of Syphonia **(8) Dissidia Final Fantasy** (9) Kingdom Hearts coded (10) The World Ends with You |
| Butterflies of Africa | (1) Neocoenyra bioculata (2) Phalanta madagascariensis (3) Mimacraea neokoton | (1) Neil Cazares-Thomas (2) Rhombophryne testudo (3) The Sick Child (4) Dillon Gordon (5) False smooth snake (6) Homoeosoma botydella (7) Northern Chinese softshel turtle (8) Catocala neogama (9) Hearths Like Ours (10) Iron Dragon |
| Giro d'Italia cyclists | (1) Paolo Savoldelli (2) Maxim Belkov (3) Pietro Bestetti | **(1) Giovanni Visconti (2) Franco Pellizotti (3) Gilberto Simoni (4) Michael Albasini (5) Paolo Bettini (6) Stefano Garzelli (7) Juan Antonio Flecha (8) Daniele Bennati** (9) Thibaut Pinot **(10) Baden Cooke** |

Figure 1: Top 10 predictions for a set of randomly selected Wikipedia, given 3 exemplars. Correct predictions are bolded.

Tables 1 and 2 show results. RELIC outperforms both the BERT name and description encoders when we restrict the candidate set to the entities seen at least 10 times in RELIC's training data, and the gap in performance increases as we increase the entity frequency threshold. However, both the name and description encoders outperform RELIC on very infrequent entities, since they can generalize from other entities with similar naming conventions or descriptions, while RELIC's embedding matrix treats every entity completely separately.

Figure 1 shows examples of predictions for Wikipedia categories given 3 exemplars for 5 randomly sampled categories. Most categories show high precision in the top 10 predictions. The category *Butterflies of Africa* fails—this is likely due to the fact that the 3 exemplars appeared only a total of 4 times in our pretraining data. The *Giro d'Italia cyclists* category is very well predicted—the single incorrect prediction *Thibaut Pinot* did cycle in the Giro d'Italia. However, for *Video games featuring female protagonists*, most of RELIC's success is due to just retrieving variations of the Final Fantasy series.

## 5 CONCLUSION

We demonstrated that the RELIC fill-in-the-blank task allows us to learn highly interesting representations of entities with their own latent ontology, which we empirically verify through a few-shot Wikipedia category reconstruction task. We encourage researchers to explore the properties of our entity representations and BERT context encoder, which we will release publicly.

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
