# OpenReview forum: "Learning Entity Representations for Few-Shot Reconstruction of Wikipedia Categories"
_ICLR.cc/2019/Workshop/LLD — LLD 2019_

### Official Review · AnonReviewer1 · 2019-04-05
**Sound approach, but the results are somewhat unconvincing**

**Rating:** 3
**Confidence:** 2

**Review:**

This paper presents an approach to obtaining representations of entities from raw text alone. They compare their method, 'RELIC', with some presumably competitive baselines, i.e., two ways of encoding entities with BERT (Devlin et al. 2018). The authors do a good job of connecting their work to previous work, noting the similarities to entity linking, entity typing, and learning entity representations from knowledge bases.

In terms of the BERT context encoder, I'm not so clear on how these representations are projected into the d-dimensional space. (I assume that d=300, as noted in 3.2?). The authors mention a transformation matrix W, but do not seem to take up how this is learned. Depending on how this is done, this could have a substantial impact on the results reported. What accuracies does BERT get without this dimensionality reduction?

The results are not altogether convincing, as the majority of entities only occur relatively rarely -- more than half have a frequency in the interval [0, 10). I interpret this as showing that the RELIC system is not particularly successful at the few-shot task, as only relatively high frequency entities have better representations with RELIC than with the BERT comparisons.

Finally, the conclusions strike me as somewhat odd. What do you mean by that RELIC "[...] allows us to learn highly interesting representations"? An added qualitative analysis of the representations could serve to highlight this further.

Minor comments:
* Footnotes should be after punctuation
* Is the 'All' row in Table 2 micro or macro averaged?

---

### Official Review · AnonReviewer2 · 2019-04-07
**Small but solid contribution**

**Rating:** 4
**Confidence:** 3

**Review:**

The paper describes a method for embeddings entities, making use of the pre-trained BERT model.
Entities are represented either as simple embeddings, by encoding the entity name with BERT, or by encoding the first paragraph of the wikipedia entry with BERT.
The contexts from entity mentions are also encoded with BERT and then optimized to be in the same space as the entity representations.
The model is evaluated on a task of producing more entities for a given category, given some examples. The proposed method performs best with more frequent entities and is outperformed on less frequent entities.

The method is fairly straightforward, with limited novelty, applying the existing BERT model to encode textual representations.
However, the paper does present a comparison and analysis for different model variations on this task, which provides some useful insight.
Also, the frequency-based analysis of the entities is interesting, showing a clear boundary where the proposed model starts outperforming the baseline.

It is unclear what is the difference between the "small number of exemplars (3-10)" and "one correct candidate entity" and how these are used differently.
Also, for experiments where the candidates do not cover all possible entities, how were the candidates chosen?

---

### Decision · Program_Chairs · 2019-04-16
**Acceptance Decision**

Accept